# Loss of *foxo* rescues stem cell aging in *Drosophila* germ line

**Filippo Artoni[1,2†], Rebecca E Kreipke[1,2†], Ondina Palmeira[1,2,3], Connor Dixon[1,2], Zachary Goldberg[1,2], Hannele Ruohola-Baker[1,2]***

[1]Department of Biochemistry, University of Washington, Seattle, United States; [2]Institute for Stem Cell and Regenerative Medicine, University of Washington, School of Medicine, Seattle, United States; [3]Nucleus of Multidisciplinary Research, Universidade Federal do Rio de Janeiro, Duque de Caxias, Brazil

**Abstract** Aging stem cells lose the capacity to properly respond to injury and regenerate their residing tissues. Here, we utilized the ability of *Drosophila melanogaster* germline stem cells (GSCs) to survive exposure to low doses of ionizing radiation (IR) as a model of adult stem cell injury and identified a regeneration defect in aging GSCs: while aging GSCs survive exposure to IR, they fail to reenter the cell cycle and regenerate the germline in a timely manner. Mechanistically, we identify *foxo* and mTOR homologue, *Tor* as important regulators of GSC quiescence following exposure to ionizing radiation. *foxo* is required for entry in quiescence, while *Tor* is essential for cell cycle reentry. Importantly, we further show that the lack of regeneration in aging germ line stem cells after IR can be rescued by loss of *foxo*.

DOI: https://doi.org/10.7554/eLife.27842.001

## Introduction

In tissues with continuous cellular turnover, homeostasis is maintained by resident populations of adult stem cells. These cells both self-renew to maintain a constant pool of pluripotent cells and differentiate into a variety of cell types to replace cells that are lost to either natural wear and tear or to acute injury and insult (*Fuchs et al., 2004*). As tissues age, the ability of adult stem cells to replenish tissues is impaired (*Schultz and Sinclair, 2016*). As a result, tissue function declines, leading to a number of different age-related deficits: grey hair is a result of impaired melanocyte maintenance (*Nishimura et al., 2005*), decreased immunity results from reduced hematopoietic stem cell populations (*Linton and Dorshkind, 2004*), and decreases in neuron production has been implicated in the pathogenesis of a number of different neurodegenerative disorders, such as Alzheimer's Disease (*Donovan et al., 2006*). However, the mechanisms that govern the regenerative competence of aging adult stem cells remain unclear. Of particular importance is the period when age-related declines first begin to manifest – when baseline stem cell function is preserved, yet, the ability to recover from injury may be impaired.

One of the most prevalent causes of injury in adult stem cells is genotoxic stress, such as that induced by exposure to ionizing radiation (IR). The fly is a particularly interesting model organism with which to examine stem cell survival post IR because recent work has demonstrated that there are several cell populations that display differing levels of resistance to ionizing radiation. Previous work in the young fly has shown a remarkable ability of *Drosophila* germline stem cells (GSCs) to survive IR, even when their progeny undergo rapid apoptosis. GSCs are resistant to the apoptotic effects of ionizing radiation (*Xing et al., 2015*): when flies are exposed to low doses of ionizing radiation GSCs survive, while their progeny, the transiently amplifying cells, do not. Dying GSC daughter cells secrete the ligand Pvf1, which signals via the Tie receptor and microRNA bantam to inhibit the apoptotic machinery in GSCs (*Bilak et al., 2014*; *Xing et al., 2015*). After a period of quiescence,

***For correspondence:**
hannele@uw.edu

[†]These authors contributed equally to this work

**Competing interests:** The authors declare that no competing interests exist.

**eLife digest** Stem cells are unspecialized cells that have the unique ability to replace dead cells and repair damaged tissues. To give rise to new cells, stem cells need to divide. This process, known as the cell cycle, includes several stages and is regulated by many different genes.

For example, in many organisms, a gene called *foxo* helps cells respond to stress and to regulate the cell cycle and cell death. Defects in this gene have been linked to age-related diseases, such as cancer and Alzheimer's disease. Previous research has shown that *foxo* can also regulate *Tor* – a gene that helps cells to divide and grow.

As we age, stem cells become less efficient at regenerating tissues, especially after exposure to toxins and radiation. However, until now, it was not known how stem cells control their division after injury and during aging, and what role these two genes play in injured and aging stem cells.

Now, Artoni, Kreipke et al. used germline stem cells from fly ovaries to investigate how young and old stem cells respond to injury. In young flies, *foxo* paused the cell cycle of the damaged stem cells. After 24 hours, *Tor* was able to overcome the action of *foxo*, and the stem cells resumed dividing and regenerating the damaged tissue. However, in old stem cells, *foxo* and *Tor* were misregulated and the stem cells could not restart dividing or repairing tissue after injury. When the levels of *foxo* in old stem cells were experimentally reduced, their ability to regenerate the tissue was restored.

These discoveries provide new insights into how stem cells respond to injury and suggest that stem cell aging may be a reversible process. A next step will be to investigate why *foxo* and *Tor* are misregulated during aging and how these two genes interact with each another. In future, this could help develop new anti-aging therapies that can restore the body's natural ability to repair itself following injury. Moreover, since cancer cells can become resistant to conventional cancer treatment by withdrawing from the cell cycle, developing new treatments that target *foxo* and *Tor* could help beat cancer and prevent its reoccurrence.

DOI: https://doi.org/10.7554/eLife.27842.002

the GSCs re-enter the cell cycle and, ultimately, regenerate the germline. Knockdown of Pvf1, a Tie ligand, in differentiating daughter cells rendered stem cells sensitive to IR, suggesting that differentiating daughter cells send survival signals to protect stem cells for future repopulation. Similar pools of IR-resistant cells have also been identified in other tissues. For example, in the larval imaginal disc, there is a population of IR-resistant cells that are able to generate viable adult tissues, even when exposed to high levels of radiation (*Verghese and Su, 2016*). Today, however, the ability of aging adult stem cells to maintain their resistance to ionizing radiation remains unexamined. Gaining a better understanding of stem cells' ability to recover from ionizing radiation will provide valuable insight into a wide range of physical phenomena, ranging from development of cancer therapeutics to improved aging remedies.

Stem cells in *Drosophila melanogaster* are a versatile system with which to study age related changes in regenerative potential (*Lucchetta and Ohlstein, 2017*; *Fabian and Brill 2012*; *Resende et al., 2017*; *Resnik-Docampo et al., 2017*). Defects in GSC function in aged flies have been identified and are in line with hypothesized defects in aging human stem cells: decreased proliferative capacity, accumulation of DNA damage, and eventual loss of stem cells (*Zhao et al., 2008*; *Kao et al., 2015*). However, the initiation of the aging process, and, particularly, how GSCs early in the aging process respond to injury, remains an open area of investigation. Furthermore, the ability of aging GSCs to regenerate their resident tissue following injury has not been fully elucidated. Since a hallmark of aging stem cells is the inability to properly regenerate tissue following injury and insult (*Sharpless and DePinho, 2007*), it is critical to understand the relationship between the initiation of aging and the ability of stem cells to recover from injury, such as following exposure to ionizing radiation.

Here, we identify and mechanistically dissect a regeneration defect in aging GSCs following exposure to ionizing radiation. Aging GSCs survive exposure to radiation, but exhibit a defect in cell cycle reentry upon completion of DNA repair. We further show that young GSCs enter a 24 hr period of quiescence following exposure to ionizing radiation before reentering the cell cycle and beginning

to regenerate the germline. In our investigation of the mechanisms governing this process, we identify the *foxo*-encoded transcription factor and the human mTOR ortholog, *Tor* as important regulators for GSC entry and exit of quiescence following exposure to ionizing radiation, respectively. Lastly, we show that the regeneration defect of aging GSCs can be rescued by knockdown of *foxo*, suggesting that misregulation of *foxo* may underlie the regenerative decline with age.

## Results

### Aging germline stem cells survive exposure to ionizing radiation, but fail to re-enter the cell cycle in a timely fashion

In the *Drosophila* ovary, at the apical tip of each germarium are two to three germline stem cells (GSCs) in direct contact with their somatic niche (Spradling 1993). These GSCs undergo asymmetric rounds of self-renewing divisions to give rise to a new stem cell and to a transiently amplifying cell (cystoblast) that undergoes four incomplete divisions generating an interconnected 16 cell cyst, of which one cell will eventually become an oocyte. Intricate interactions between GSCs and somatic cells allow for GSC maintenance in the niche (*Fuchs et al., 2004*; *Ward et al., 2006*). Germline stem cells can be identified by their proximity to the cap cells in the niche and the prominent foci of adducin staining, labeling the subcellular structures called spectrosomes, while progeny can be identified by a branched focus of adducin, known as the fusome, in cells that do not reside within the niche (*Figure 1A*.). As GSCs progress through the cell cycle, they alternately have an elongated or a round spectrosome, the morphology of which can be used to identify dividing GSCs (*Figure 1A*, (*de Cuevas and Spradling, 1998*). GSCs in the female *Drosophila* ovariole lose replicative capacity with age (*Pan et al., 2007*; *Zhao et al., 2008*; *Kao et al., 2015*), however, the early steps in aging and GSC ability to survive exposure to ionizing radiation during the early aging process has not been probed.

We first asked whether aging GSCs survive radiation exposure. Since it has been shown that young GSCs survive exposure to IR and are able to regenerate the germline by one week following exposure to IR (*Xing et al., 2015*), we probed the system to identify the earliest time points where we could observe a defect in the aging GSCs' ability to recover from exposure to IR. We found that at 4 weeks, recovery from IR and regeneration was normal, while a defect could be observed when 6 week old animals were irradiated. We exposed 4- and 6 week old wild type flies to 50 Grays of radiation and quantified the number of GSCs in unirradiated flies and compared them to the number of GSCs in germaria of flies one week following irradiation. We found that, although 4 and 6 week old germaria lose a small number of GSCs one week post IR the majority of 4 and 6 week old germaria still had 1 to 2 GSCs one week following irradiation (*Figure 1B*), indicating that the Tie-mediated protective mechanism remains mainly intact in aging germaria. Next, we assayed the level to which the GSCs were able to regenerate the germarium following exposure to ionizing radiation. We visualized GSC progeny with adducin staining and compared the number of germaria that had four or more progeny to those that had fewer than four progeny in unirradiated flies and in germaria of flies one week following exposure to irradiation. We found that, while at 4 weeks, the number of germaria with progeny was not significantly different before and after exposure to irradiation, at 6 weeks, the number of germaria with progeny one week following irradiation was significantly lower than in the germaria of unirradiated flies (*Figure 1C*). This suggests that, while aging GSCs are able to survive exposure to irradiation, they are unable to re-enter the cell cycle and regenerate the germarium in a timely manner. We confirmed that regeneration was impaired by assaying the percentage of GSCs with elongated spectrosomes, which is an indication of GSC division. We found that levels of spectrosome elongation were similar before and after irradiation in the 4 week old animals, however, in the 6 week old animals, there was a significant decrease in the percentage of GSCs with elongated spectrosomes one week after irradiation (*Figure 1D*). We further confirmed that regeneration was impaired in 6 week old animals by comparing the number of adults produced by 4 and 6 week old irradiated animals. We found that while irradiated 4 week old animals produced less adults than unirradiated animals of the same age , this defect was much more pronounced in irradiated 6 week old animals (*Figure 1—figure supplement 1A,B*). Taken together, our data suggest that aging GSCs are able to survive exposure to low IR, but are unable to reenter the cell cycle and regenerate the germline.

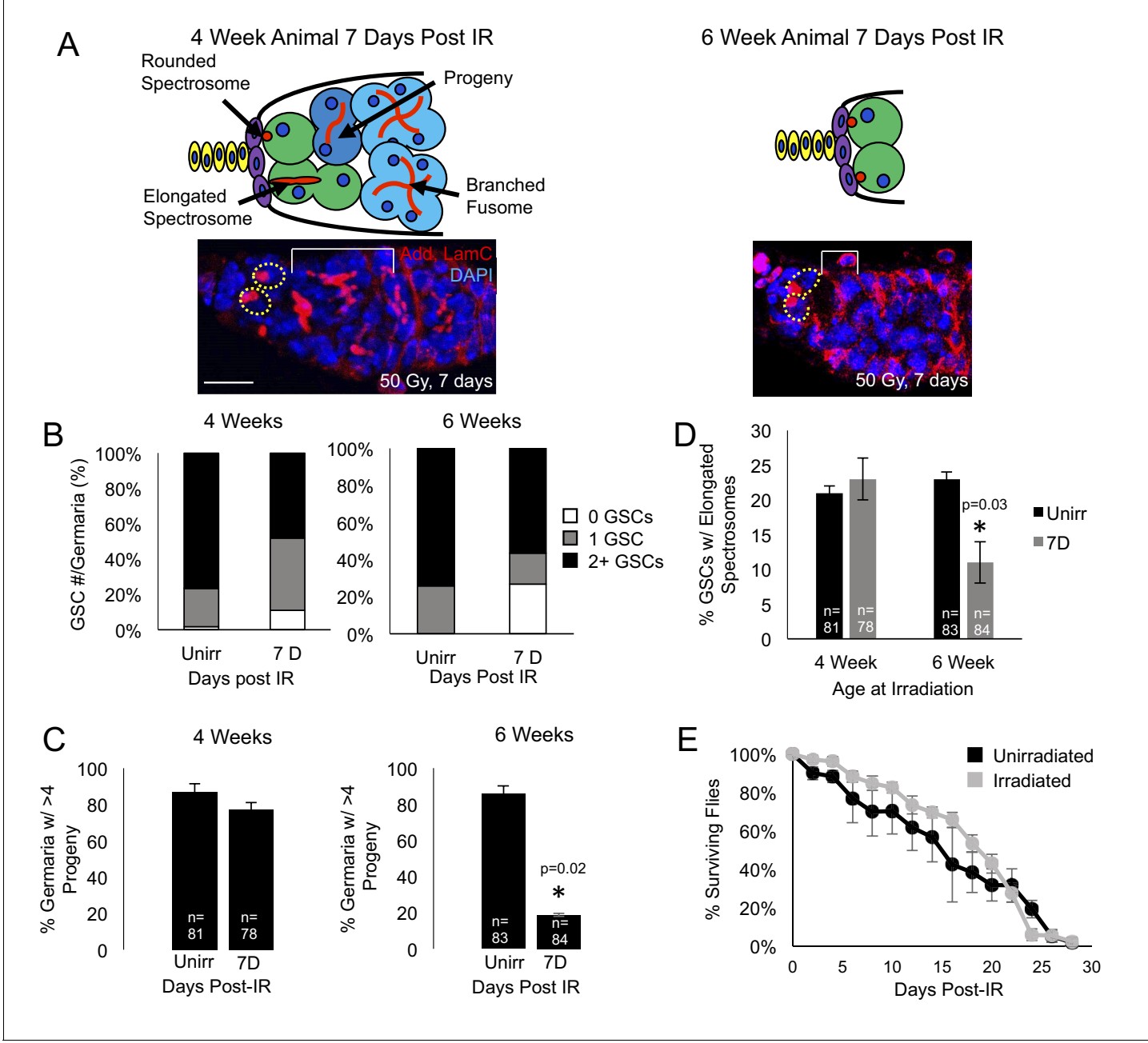

**Figure 1.** Aging GSCs survive exposure to IR, but do not regenerate the germarium. (**A**) Top: Schematic diagrams of 4 week old vs. 6 week old germaria one week following IR. Terminal filament cells, yellow; cap cells, purple; germline stem cells, green, cystoblasts, dark blue; cysts, light blue. Bottom: representative images of 4 week old (left) and 6 week old (right) germaria stained for adducin and laminC (red) and DAPI. Germline stem cells (dotted yellow line) are visible in both 4 week old and 6 week old germaria, however, branched fusomes are only seen in the 4 week old germaria (white bracket). Scale bar = 10 μm. (**B**) Quantification of the number of GSCs/germaria in 4 week old (left) and 6 week old (right) germaria, before and one week following IR. White, 0 GSCs, grey, 1 GSC, black, 2 + GSCs. (**C**) Quantification of the percent of germaria with 4 or more progeny in 4 week old (left) and 6 week old (right) flies before and after IR. In 4 week old animals, there is no significant difference in the percent of germaria with 4 or more progeny, while in 6 week old animals, there is a decline in the percentage of germaria with 4 or more progeny after IR. (**D**) Quantification of the percentage of GSCs with elongated spectrosomes in 4 week old (left) and 6 week old (right) animals. While there is not a significant difference in the percentage of elongated spectrosomes before and one week following IR in 4 week old animals, there is a decrease in the percentage of GSCs with elongated spectrosomes one week following IR in 6 week old animals. (**E**) Survival curve of animals after 6 weeks, comparing unirradiated (black) and irradiated (grey) flies. There was no significant difference in the survival time of animals between irradiated and unirradiated flies.

DOI: https://doi.org/10.7554/eLife.27842.003

The following figure supplement is available for figure 1:

*Figure 1 continued on next page*

*Figure 1 continued*

**Figure supplement 1.** Decreased fertility following irradiation in old flies.

DOI: https://doi.org/10.7554/eLife.27842.004

To assay the general fitness of the 6 week old flies, we compared the survival rates of 6 week old flies, both irradiated and unirradiated. We found no significant difference in the life span of the irradiated and unirradiated flies (*Figure 1E*). This indicates that we have found a time when GSCs have begun to age and show deficits in regenerative capacity, but dramatic aging phenotypes are not yet detectable at the organismal level. Hence, our analysis will allow us to understand the earliest processes in stem cell aging.

## DNA damage is repaired within 24 hr following exposure to IR in aging animals

DNA damage can inhibit cell cycle progression (*Bunz et al., 1998*; *Reinhardt and Schumacher, 2012*), and increases in levels of DNA damage have been reported in aged GSCs (*Kao et al., 2015*). To assay whether the observed delay of cell cycle reentry in aging GSCs following irradiation was due to delays in DNA damage repair, we exposed 6 week old flies to 50 Gys of ionizing radiation. We then dissected ovaries from flies 30 min, 24 hr, and 7 days following irradiation and compared levels of DNA damage in GSCs to those of unirradiated flies, as visualized by γH2AV staining (*Figure 2A–C*). We compared the number of GSCs with high, moderate, or minimal levels of DNA damage at these time points. We found that DNA damage peaked 30 min following IR, with a majority of GSCs showing high levels of γH2AV staining (*Figure 2B,D*). However, by 24 hr following IR, levels of DNA damage had returned to baseline levels, similar to those in unirradiated flies (*Figure 2C–E*). Additionally, by 7 days post-irradiation, there was no significant difference in the level of DNA damage compared to unirradiated flies (*Figure 2D*). This indicates that DNA damage repair has concluded, even though the aging GSCs remain unable to regenerate the germline, suggesting that additional mechanisms must be responsible for the aging defect we identified in 6 week old GSCs.

## Young germline stem cells enter a brief period of quiescence following exposure to IR

Having identified a regeneration defect in aging GSCs, we next investigated the timing of IR induced cell cycle exit and reentry in young, healthy flies. We exposed 2–7 day old flies to 50 Gys of ionizing radiation and compared levels of GSC division and regeneration to unirradiated flies at 24 hr intervals. We visualized branched fusomes and spectrosomes via adducin staining (*Figure 3B–E*). In order to assay the rates of GSC division, we compared the morphology of the spectrosomes in GSCs from flies that had been irradiated to unirradiated flies. We quantified the percentage of GSCs with elongated spectrosomes, as an indicator of GSC cellular division (*Figure 3B,D*, yellow arrow). We observed a significant decrease in the percentage of GSCs with elongated spectrosomes one day post-IR (*Figure 3F*). By two days following irradiation, the percentage of GSCs with elongated spectrosomes had returned to baseline (*Figure 3F*). This suggests that when well fed, young animals are exposed to low doses of irradiation, GSCs enter a brief, approximately 24 hr period of quiescence.

Similarly, we quantified the number of regenerated germaria by quantifying the number of germaria with germ line cysts containing branched fusomes (*Figure 3G*). Unlike GSCs, transiently amplifying cells do not survive exposure to ionizing radiation and the number of new daughter cells can, therefore, be used as an indirect measure of GSCs' regeneration capacity following irradiation damage (*Xing et al., 2015*). There was a significant decline in the percentage of germaria with GSC daughters containing branched fusomes one and two days post-IR (*Figure 3G*). By 3 days, post-IR, the percentage of regenerated germaria had dramatically increased, with complete recovery achieved by 4 days post-IR. This suggests that GSCs give rise to progeny by 3 days post-IR, which is in line with our observation that GSCs begin dividing around two days post-IR. Taken together, our data indicate that GSCs enter an approximately 24 hr period of quiescence after exposure to ionizing radiation before returning to the cell cycle and regenerating the germline.

*Notch* signaling also plays an essential role in the development and maintenance of the *Drosophila* germline stem cell niche. Niche cells and GSCs communicate with one another via the Delta and

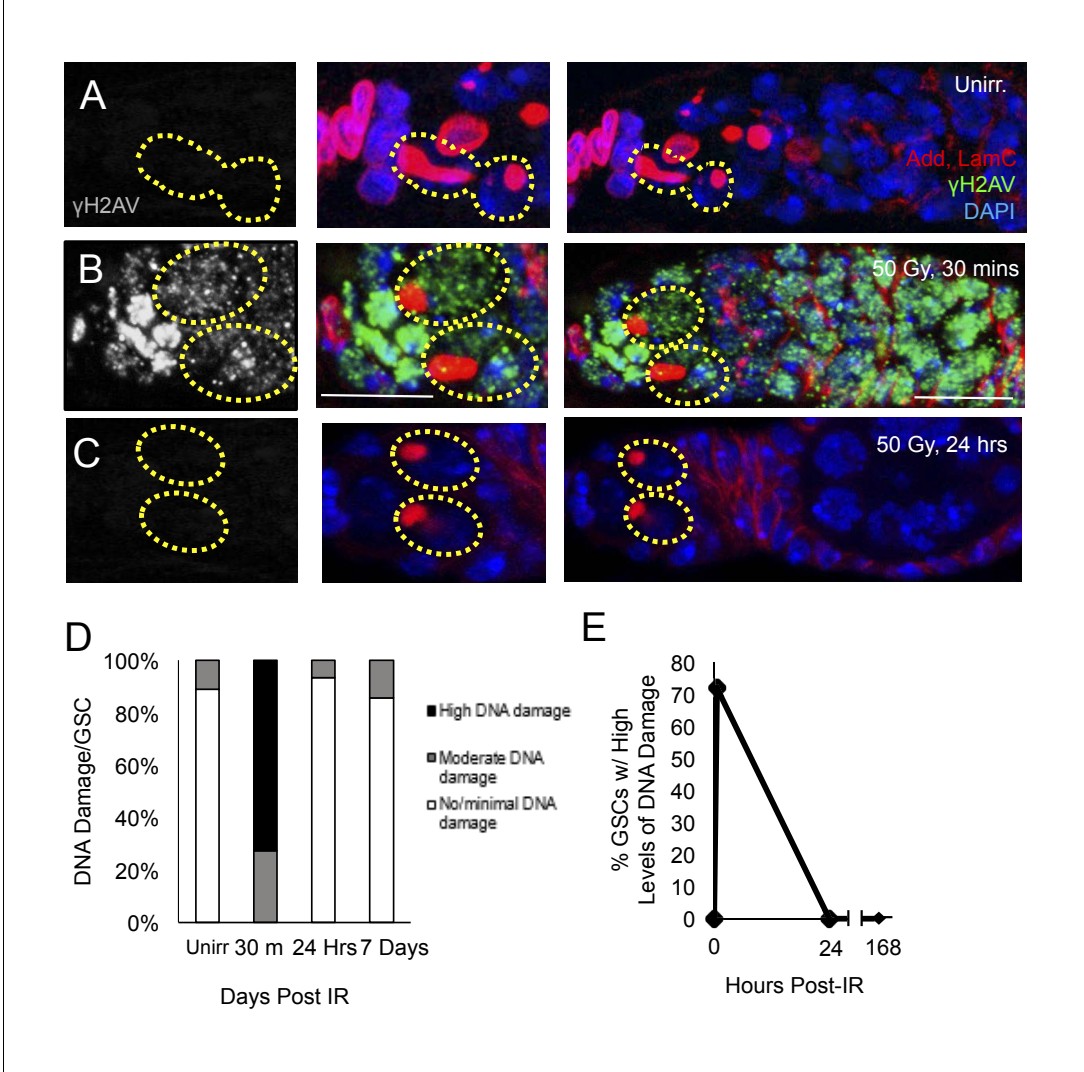

**Figure 2.** DNA damage repair concludes within 24 hr in aging animals. (A-C) Representative images of a 6 week old germaria stained for adducin and lamC (red), γH2AV (green), and DAPI. (A) Unirradiated 6 week old germaria showing examples of GSCs with no γH2AV staining. (B) 6 week old germaria, 30 min post-IR. Scale bar = 10 μm. (C) 6 week old germaria, 24 hr post- IR. (D) Stacked bar plot showing percentage of GSCs with low (white), medium (grey), or high (black) levels of γH2AV staining following IR. High levels of DNA damage peak 30 min following IR and return to baseline by 24 hr. (E) Line graph showing percentage of GSCs with high levels of γH2AV staining over time.

DOI: https://doi.org/10.7554/eLife.27842.005

Serrate Notch ligands to regulate various niche features, including niche size and GSC number (*Ward et al., 2006*; *Song et al., 2007*). Abrogation of *Notch* signaling by expressing a nos-Gal4-inducible RNAi construct against *neuralized* (neur), a ubiquitin ligase which mediates the internalization and subsequent activation of the Delta and Serrate Notch ligands in the germline, resulted in a complete loss of GSCs, even before exposure to ionizing radiation (data not shown), confirming the essential role of *Notch* signaling in GSC maintenance (*Ward et al., 2006*; *Song et al., 2007*). To study if supernumerary GSCs follow the wild type GSC kinetics of post-IR quiescence, we drove overexpression of *Delta* in the germline using the Gal4 system. Nos-Gal4 > Delta germaria showed an increased number of spectrosome marked cells, which we confirmed were GSCs via expression of the TGFβ target, *Dad* (*Figure 3H–I*). This indicates that the expanded *TGFβ* signaling from niche induced extranumerary GSCs, as seen previously (*Ward et al., 2006*; *Song et al., 2007*). We exposed *Delta* overexpression flies to ionizing radiation (50 Gys) and dissected their ovaries 1, 2 and

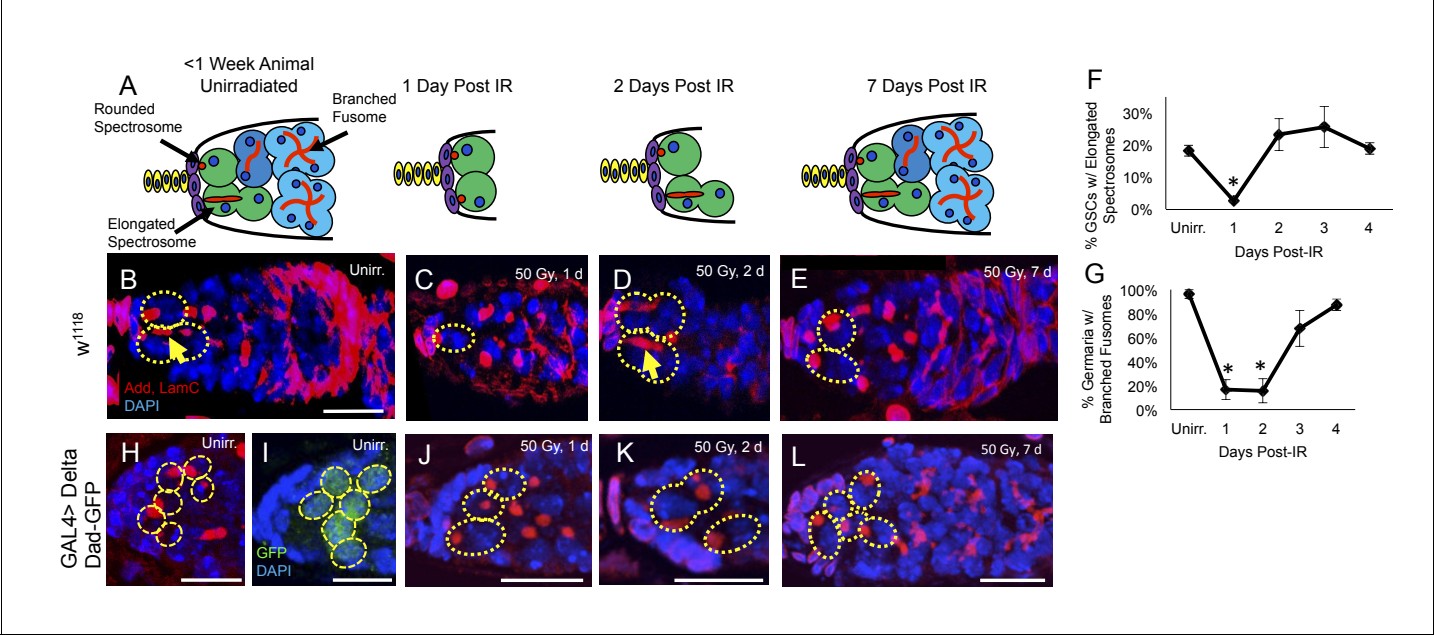

**Figure 3.** Young GSCs enter a brief period of IR-induced quiescence before returning to the cell cycle. (**A**) Schematic of the progression of cell loss and recovery following exposure to IR in young animals. Terminal filament cells, yellow; cap cells, purple; germline stem cells, green, cystoblasts, dark blue; cysts, light blue. (**B-E**) Representative images of young w[1118] germaria stained for adducin and lamC (red) and DAPI (blue). GSCs are indicated by the presence of a spectrosome and DAPI staining (dotted yellow line); elongated spectrosomes indicated with yellow arrow. Scale bar = 10 μm. B. Unirradiated germarium. (**C**) Germarium 1 day post-IR (50 Gγs). (**D**) Germarium 2 days post-IR. (**E**) Germarium 7 days post-IR. (**F**) Line graph of percentage of GSCs with elongated spectrosomes for days 1–4 post-IR. There is a significant decrease of percentage of GSCs with elongated spectrosomes at 1 day post-IR (3 biological experiments, mean ± s.e.m., *p<0.05, ANOVA). (**G**) Line graph of the percentage of germaria with branched fusome for days 1-4 post IR. (**H**) Representative unirradiated germaria of nos-Gal4 > *Delta* flies, showing increased niche size and supernumerary GSCs. (**I**) Unirradiated nos-Gal4 > *Delta; Dad*-GFP germaria. (**J**) nos-Gal4 > Delta germaria 1 day post-IR. (**K**) nos-Gal4 > *Delta* germaria 2 days post-IR. (**L**) nos-Gal4 > *Delta* germaria one week post-IR showing a fully regenerated germarium.

DOI: https://doi.org/10.7554/eLife.27842.006

The following figure supplement is available for figure 3:

**Figure supplement 1.** Extranumerary GSCs in Delta overexpression are reduced after IR.

DOI: https://doi.org/10.7554/eLife.27842.007

7 days post IR. GSCs in the expanded niche enter and exit quiescence in a timely manner (*Figure 3H–L*). However, while the somatic niche remained large, GSC number was reduced one day after IR (*Figure 3—figure supplement 1*), suggesting that the protective signal from daughter cells cannot penetrate to protect all the supernumerary GSCs after exposure to ionizing radiation.

## Germline stem cell DNA damage is repaired within 24 hr of exposure to IR

To confirm that DNA damage repair kinetics are not substantially different in young flies and old flies, we assayed levels of DNA damage via γH2AV staining in the germaria of 2–7 day old flies exposed to ionizing radiation (*Figure 4A–H*). We quantified the percentage of GSCs with high, moderate, or no/minimal levels of γH2AV at 30 min, 12 hr, and 24 hr after exposure to ionizing radiation and compared this to levels of γH2AV in unirradiated germaria (*Figure 4I*). We found that there was a significant increase in the percentage of GSCs with high levels of DNA damage 30 min post-IR (*Figure 4C,D,I*). By 12 hr post-IR, a majority of the germaria had repaired DNA damage to a moderate amount: only 8% showed high levels of DNA damage, while 83% had moderate levels of DNA damage (*Figure 4E–F,I–J*). By 1 day following radiation exposure, only 34% of GSCs had moderate levels of yH2AV staining, with 66% of GSCs returned to baseline levels of DNA damage (*Figure 4G, H,I,J*). This suggests that DNA damage repair kinetics in young flies resemble those of the aging fly,

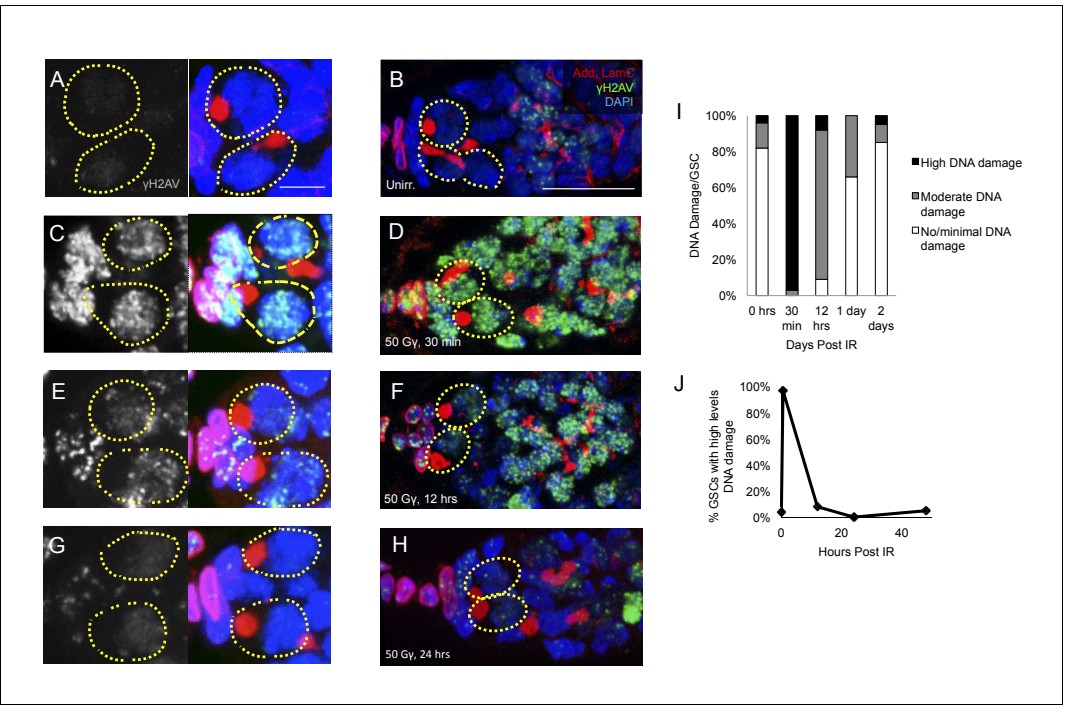

**Figure 4.** DNA damage repair concludes by 24 hr in young flies A-H.  Representative images of young germaria stained for γH2AV alone (grey) or adducin/lamC (red), γH2AV (green), and DAPI. (**A**) Unirradiated GSCs. γH2AV channel alone (left), color merge (right). Scale bar = 5 μm (**B**) Unirradiated germarium. Scale bar = 10 μm (**C**) GSCs 30 min post-IR. γH2AV channel alone (left); color merge (right) (**D**) Germarium 30 min post-IR. (**E**) GSCs 12 hr post-IR. γH2AV channel alone (left); color merge (right). (**F**) Germarium 12 hr post-IR. (**G**). GSCs 24 hr post-IR. γH2AV channel alone (left); color merge (right). (**H**) Germarium 24 hr post-IR. (**I**) Stacked bar plot showing percentage of GSCs with low (white), medium (grey), or high (black) levels of γH2AV staining following IR. High levels of DNA damage peak 30 min following IR and return to baseline by 24 hr. (**J**) Line graph showing percentage of GSCs with high levels of γH2AV staining over time.

DOI: https://doi.org/10.7554/eLife.27842.008

supporting our previous findings that alterations in DNA damage repair kinetics alone cannot account for the regeneration defect in aging GSCs.

We next asked what mechanisms are involved in regulating IR-induced quiescence in GSCs. We first probed the role of the G1 checkpoint in IR-induced quiescence by manipulating levels of the p21 ortholog, *dacapo*. We found that, while overexpression of *dacapo* was sufficient to prolong IR-induced quiescence (*Figure 5—figure supplement 1B*), there was no significant difference in the ability of GSCs to enter quiescence when *dacapo* levels were reduced (*Figure 5—figure supplement 1E,F*). This suggests that *dacapo* is not required for GSCs to enter quiescence after a radiation challenge, suggesting that the G1 checkpoint is not where GSCs arrest following exposure to IR. Additionally, we examined the role of the DNA damage sensing machinery in regulating IR-induced quiescence. We found that when the CHK2 ortholog, *loki*, was knocked down via RNAi, it impaired the ability of GSCs to enter quiescence (*Figure 5—figure supplement 1G and H*), consistent with recent work demonstrating the vital role of *loki* in regulating GSC survival following exposure to high levels of IR (*Ma et al., 2016*). Thus, we worked to identify the functional machinery downstream of CHK2 that regulates the stem cell quiescence.

## *foxo* is required for GSC cell cycle arrest following exposure to IR

*foxo* is a key player in the cellular response to IR (*Chung et al., 2012*; *Xing et al., 2015*). To probe whether *foxo* was required for IR-induced GSC cell cycle exit and reentry, we knocked down *foxo* in the germline by crossing UAS-Dcr-2; nos-Gal4 flies to two independent UASp-*foxo* RNAi lines. We then assayed the morphology of GSCs' spectrosomes at 1 and 2 days post-IR in nos-GAL4 > *foxo*

RNAi flies and compared them to unirradiated GSCs. We found that, while in UAS-Dcr-2; nos-Gal4 control flies, there is a dramatic decrease in the percentage of GSCs with elongated spectrosomes 1 day post-IR, *foxo* deficient GSCs in both RNAi lines kept dividing at a normal rate (*Figure 5A,B*). Additionally, the percentage of germaria with branched fusomes 1 day post-IR is increased in *foxo* RNAi flies (*Figure 5C*; *Supplementary file 1A,B*), further strengthening our finding that knockdown of *foxo* eliminates IR-induced quiescence.

This suggests that *foxo* is required for GSCs to initiate IR-induced quiescence and withdraw from the cell cycle.

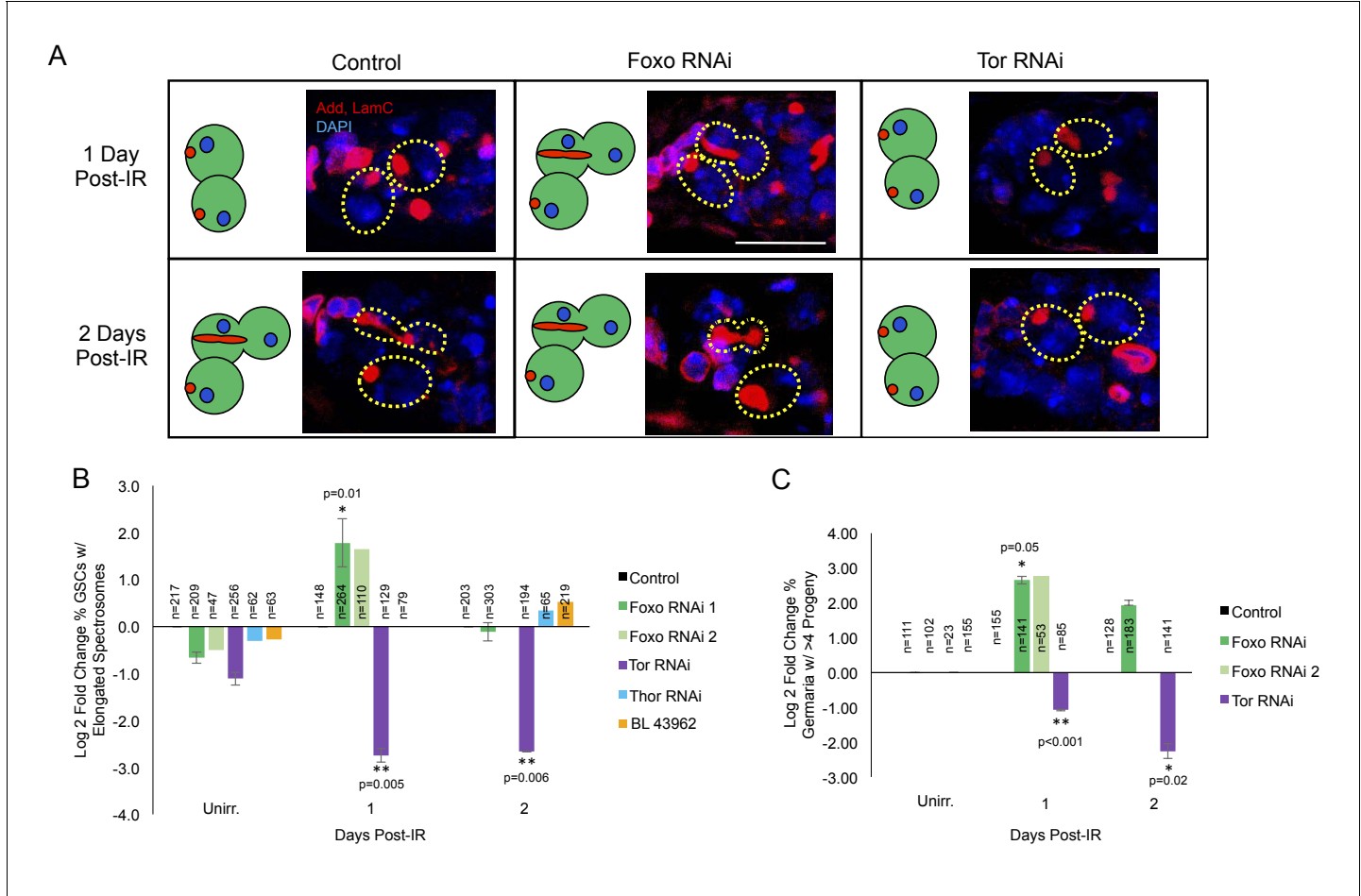

**Figure 5.** *foxo* and *Tor* regulate GSCs' cell cycle exit and reentry. (**A**) Representative images of germaria stained for adducin and lamC (red) and DAPI (blue). Left top: Control germaria, 1 day post-IR. Left bottom: Control germaria, 2 days post-IR. Middle top: *foxo* RNAi germaria, 1 day post-IR. Scale bar = 10 μm Middle bottom: *foxo* RNAi germaria, 2 days post-IR. Right top: *Tor* RNAi germaria, 1 day post-IR. Right bottom: *Tor* RNAi germaria, 2 days post-IR. (**B**) Bar plot of the percentage of GSCs with elongated spectrosomes up to two days post-IR for *foxo*, *Tor*, *Thor* and control RNAi lines, plotted as log2 fold change compared to control. *foxo* RNAi inhibits the ability of GSCs to exit the cell cycle. *Tor* RNAi inhibits the ability of GSCs to reenter the cell cycle. (**C**) Bar plot of the percentage of germaria with more than four progeny up to two days post IR for *foxo*, *Tor*, *Thor* and control RNAi lines, plotted as log2 fold change compared to control.

DOI: https://doi.org/10.7554/eLife.27842.009

The following figure supplements are available for figure 5:

**Figure supplement 1.** IR-induced quiescence occurs at the G2 checkpoint.
DOI: https://doi.org/10.7554/eLife.27842.010

**Figure supplement 2.** Loss of *foxo* radiosensitizes GSCs.
DOI: https://doi.org/10.7554/eLife.27842.011

## *Tor* is required for GSC cell cycle reentry post-IR

Foxo has been shown to regulate *Tor* in *C.elegans*, *Drosophila,* and mammalian systems (*Puig et al., 2003*; *Jia et al., 2004*; *Chen et al., 2010*). Since *Tor* signaling is known to modulate both *Drosophila* longevity and GSC division (*Bjedov et al., 2010*; *LaFever et al., 2010*) we analyzed its potential role in cell cycle regulation following exposure to IR. To probe whether *Tor* is required for IR-induced GSC cell cycle exit or reentry, we knocked down *Tor* in the germline using a nos-GAL4 driver to express a *Tor* RNAi construct under UAS control (nos-GAL4 > *Tor* RNAi). We then assayed *Tor* mutant GSC division capacity by analyzing the morphology of spectrosomes and the number of daughters produced at 1 and 2 days post-IR and compared them to control and unirradiated GSCs. We found that when *Tor* is knocked down, there is an even larger decrease in the percent of GSCs with elongated spectrosomes one day post-IR than in control animals, suggesting a higher pene-trance in cell cycle exit (*Figure 5A,B*). Furthermore, the percentage of *Tor* RNAi GSCs with elon-gated spectrosomes and the number of GSC daughters remained decreased two days post-IR, when control GSCs have reentered the cell cycle (*Figure 5C*; *Supplementary file 1A,B*), suggesting a dra-matic delay in the reentry to the self-renewing cell cycle and regenerative capacity in *Tor* mutant GSCs.

We also probed the role of *Tor* signaling pharmacologically with rapamycin. Rapamycin is a potent inhibitor of the TORC1 complex, preventing phosphorylation of Tor's downstream targets (*Sabatini et al., 1995*). Following irradiation, wild type flies were fed grape juice with either rapamy-cin (200 µM) or vehicle for two days. There was a significant decrease in the percentage of GSCs with elongated spectrosomes 2 days post-IR with rapamycin treatment (*Figure 6A–C*). Taken together, these data suggest that *Tor* is required for GSC exit from quiescence and cell cycle reentry post-IR.

Finally, we probed the question of whether IR-induced quiescence is protective to GSCs. When nos-Gal4 > *foxo* RNAi flies were exposed to a secondary dose of ionizing radiation 24 hr following the initial dose (*Figure 5—figure supplement 2A*), we found that there was a decrease in the num-ber of GSCs per germaria in *foxo* RNAi flies (*Figure 5—figure supplement 2C*). This difference can-not be attributed to *foxo* reduction alone, since unirradiated nos-Gal4 > *foxo* RNAi ovaries have a normal number of GSCs per germarium. This suggests that *foxo*-mediated IR-induced quiescence is important for GSC survival.

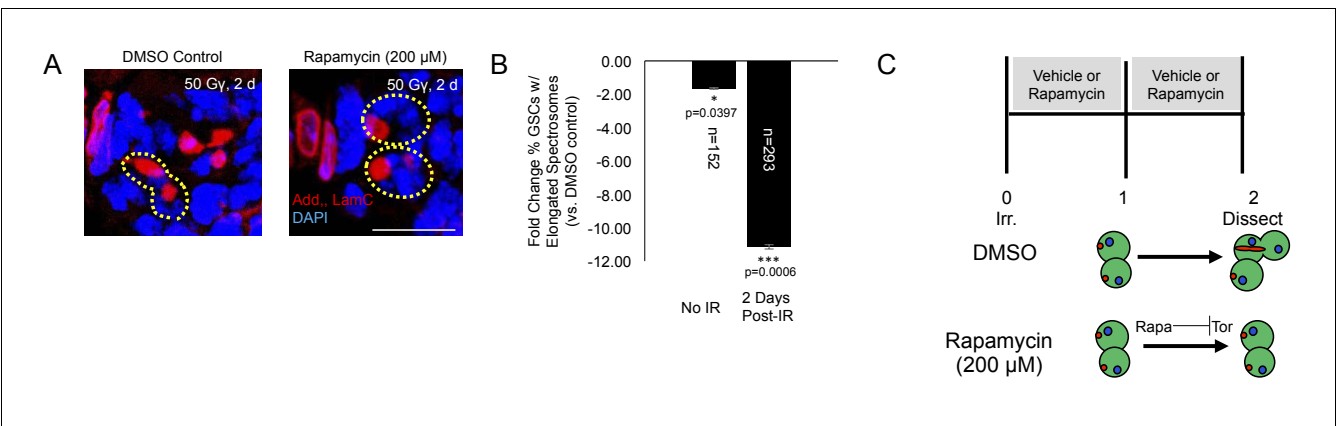

**Figure 6.** *Tor* activity is required for cell cycle reentry following IR. (**A**) Representative images of GSCs from a fly fed vehicle control (left) or rapamycin (200 µM, right) stained for adducin/lamC (red) and DAPI (blue) two days post-IR. Scale bar = 10 µm (**B**) Bar plot showing percentage of GSCs with elongated spectrosomes two days post IR. Flies fed rapamycin had decreased rates of GSC division compared to flies fed vehicle control. The effect of rapamycin on GSC division was much more pronounced two days post IR. (**C**) Top:Experimental paradigm. Flies were irradiated at Day 0 and fed either vehicle control or Rapamycin for 48 hr post-IR. Ovaries were dissected 2 days post-IR and analyzed. Bottom: When rapamycin represses Tor activity, there is a decrease in the ability of GSCs to exit quiescence.

DOI: https://doi.org/10.7554/eLife.27842.012

### *foxo* represses *Tor* in GSC after IR

Since we identified opposing roles for *foxo* and *Tor* in regulating IR-induced quiescence, we next asked whether these two signaling components operated independently or in conjunction with each other. To visualize *foxo* activity, we stained for Foxo protein and to assay levels of *Tor* activity, we stained for phosphorylated ribosomal protein S6 (p-S6), a downstream effector of TORC1. We compared levels of Foxo and p-S6 staining in young, wild type files following exposure to ionizing radiation. We observed a dramatic increase in the level of Foxo in GSCs' nuclei 1 day post-IR (*Figure 7A*). Levels of Foxo staining returned to baseline (*Figure 7D*) by 2 days post-IR. Phospho-S6 staining showed a complimentary pattern to Foxo staining: while *foxo* is highly expressed at the anterior tip of germaria and the GSCs, p-S6 levels are high in 8- and 16 cell cysts towards the posterior end of germaria, suggesting a possible regulatory role of *Tor* activity by *foxo* (*Figure 7B*). To test this, we reduced *foxo* levels and measured *Tor* activity by analyzing p-S6 patterns. When *foxo* is depleted via nos-Gal4-induced RNAi, the level of p-S6 staining increases and is observed closer to the anterior tip of the germaria and GSCs, which is not observed in wild type animals (*Figure 7C*). p-S6 staining was completely absent in germaria of nos-Gal4 > *Tor* RNAi flies (*Figure 7F*), confirming that p-S6 is a reliable measure of *Tor* activity. Together, this suggests that Tor and Foxo activity are spatially segregated due to an antagonistic relationship between the activity of these two proteins

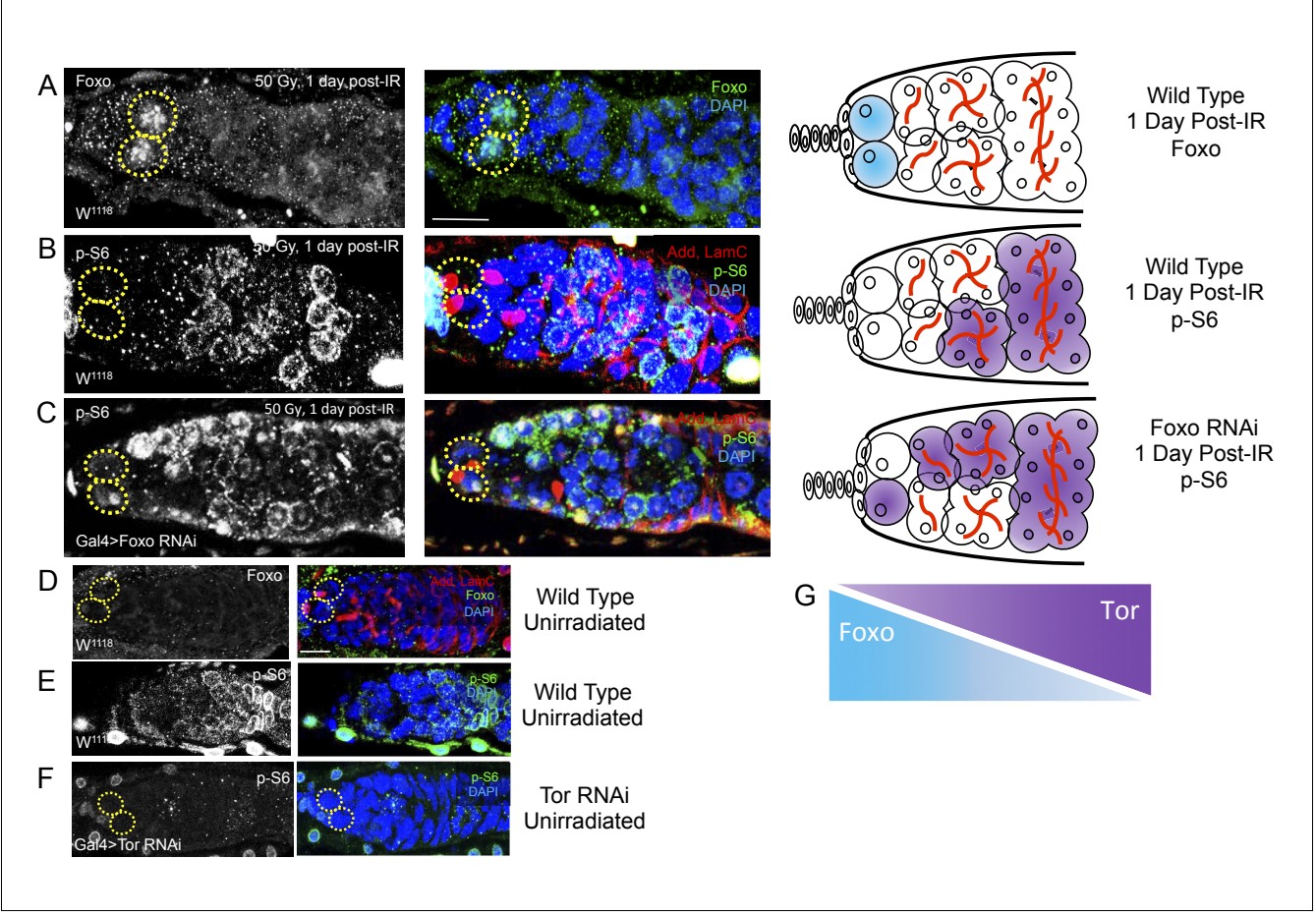

**Figure 7.** Foxo and Tor activity are spatially segregated. (**A**) Foxo levels increase in GSCs one day post-IR. Scale bar = 10 μm (**B**) Phosphorylated S6 (p–S6) and Foxo signals have opposite gradients throughout the germarium. Levels of p-S6 increase in the anterior region of the germarium 1 day post-IR, and signaling returns to baseline at 2 days post-IR. (**C**) P-S6 staining moves closer to the anterior tip of the germaria in Foxo deficient flies 1 day post-IR. (**D**) Levels of Foxo are low in wild type, unirradiated flies. Scale bar = 10 μm (**E**) p-S6 staining is localized to the posterior end of the germarium in wild type, unirradiated flies. (**F**) In *Tor* RNAi flies, there was no detectable germline p-S6 staining. (**G**) Diagrammatic representation of Foxo and Tor gradients throughout the germarium. Foxo and Tor levels are elevated, respectively, in the anterior and posterior region of the germarium.
DOI: https://doi.org/10.7554/eLife.27842.013

(*Figure 7G*). In particular, these data show that *foxo* can repress the TORC1 target, p-S6 in the *Drosophila* ovary.

## Knockdown of *foxo* levels rescues the GSC regeneration defect in aging animals

Since we identified *foxo* as a critical regulator of IR- quiescence, we next asked whether knockdown of *foxo* in the aging GSC could rescue the observed aging regeneration defect. We aged nos-Gal4 > *foxo* RNAi flies to 6 weeks and exposed them to 50 Gys of ionizing radiation. We quantified the number of GSCs per germaria, as well as the number of germaria with four or greater progeny in unirradiated and one week post-IR flies. We found that, compared to unirradiated flies, there was no significant difference in the number of GSCs/germaria in 6 week old nos-Gal4 > *foxo* RNAi flies one week following irradiation (*Figure 8A,C*). Strikingly, we also found that one week following exposure to IR, nos-Gal4 > *foxo* RNAi flies showed evidence of germline regeneration, with equal numbers of germaria with greater than four progeny when compared to their unirradiated counterparts (*Figure 8B,D*). We also observed large 8 cell cysts one week post-IR in 6 week old nos-Gal4 > *foxo* RNAi flies (*Figure 8B*) indicating robust and extensive regeneration of the germline. This developmental stage is never observed in 6 week-old wild type flies one week post-IR. These findings suggest that knockdown of *foxo* is sufficient to relieve the aging regeneration defect: aging flies with reduced levels of *foxo* are able to regenerate the germline within a week, while wild type flies cannot (*Figure 8F*).

To study *foxo*'s mode of function in the context of aging, we probed *Tor* signaling, a Foxo target repressed post-injury in young animals. Aging flies expressing a UASp RNAi construct against *foxo* showed a dramatic increase in germline *Tor* activity, as measured by p-S6 antibody staining (*Figure 8E*). This suggests that Foxo represses *Tor* activity during aging and that overactivation of Foxo may account for the inability of aging GSCs to regenerate following exposure to IR (*Figure 9*), as evidenced by the ability of the aging germline to regenerate with decreased levels of Foxo.

## Discussion

Adult stem cells experience a decrease in regenerative potential with age that results in a decrease in the ability of adult tissues to repair themselves following injury or insult. We have now identified the earliest time at which aging *Drosophila* germline stem cells lose the ability to appropriately recover from exposure to sublethal doses of ionizing radiation (IR) and dissect the mechanism for this process. Following exposure to IR, most aging GSCs survive, but fail to reenter the cell cycle and regenerate the germline, a process that is activated in young flies post IR. This is not due to a defect in DNA damage repair, as DNA damage repair concludes in a timely manner, even though the aging GSCs fail to return to the cell cycle. We have now identified two key regulators for IR induced quiescence: *foxo* and *Tor*. These two genes have opposing roles in regulating GSC cell cycle, exit and reentry after IR, respectively. Furthermore, *Tor* inactivation by RNAi or Rapamycin treatment induces a premature GSC aging phenotype, impairing *Tor*-dependent regeneration post injury. Conversely, knocking down *foxo* in aging animals rescues the aging phenotype, allowing GSCs to regenerate the germline, as observed in young flies. Finally, we show that *foxo* and *Tor* have opposing patterns of expression in the germarium and depletion of *foxo* leads to increases in *Tor* activity. This suggests that *foxo* regulates post-IR quiescence and cell cycle reentry by regulating *Tor* activity. Importantly, we show that loss of *foxo* rescues the GSC age-related regeneration phenotype due to IR. Overall, this study shows that IR induced quiescence is regulated by *foxo* and the mTOR ortholog, *Tor*, and suggests that upregulation of *foxo* and misregulation of *Tor* signaling in aging adult stem cells may be responsible for the decline in regenerative capacity following injury or insult (*Figure 9*).

Aging adult stem cells are unable to regenerate injured tissue as effectively and efficiently as young stem cells (*Schultz and Sinclair, 2016*). However, it has remained an open area of investigation as to whether this is due to a loss of adult stem cells with age or whether this is due to a decrease in the ability of adult stem cells to regenerate appropriately. Our work shows that the anti-apoptotic protective mechanisms (*Xing et al., 2015*) that shield adult stem cells from death remain mainly intact, but the aging GSCs are unable to reenter the cell cycle following IR-induced quiescence.

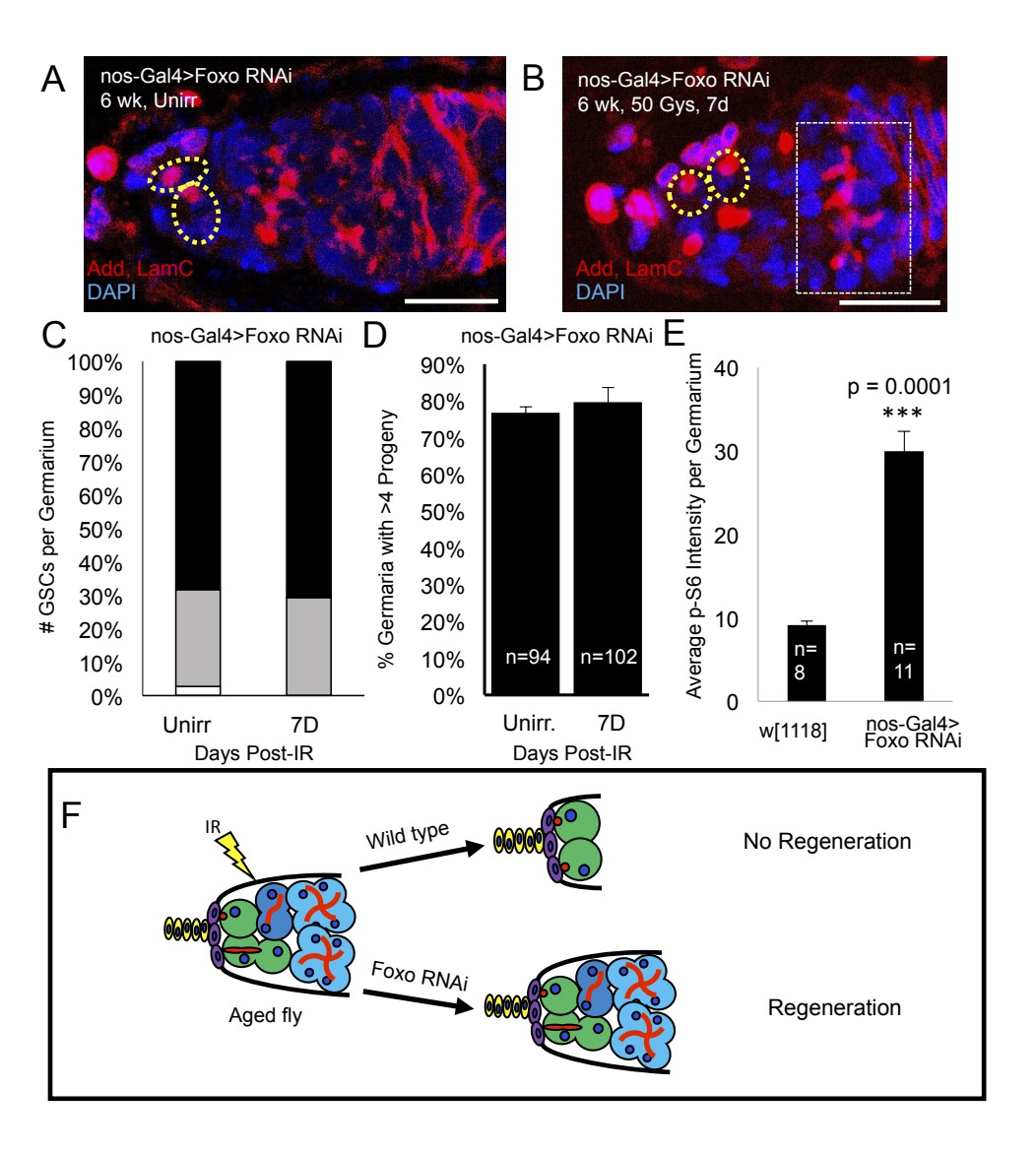

**Figure 8.** Loss of *foxo* rescues age-related regeneration defect. (**A**) Representative image of a 6 week old nos-Gal4 > *foxo* RNAi germarium. (**B**) Representative image of a 6 week old nos-Gal4 > *foxo* RNAi germarium one week post-IR, showing a fully regenerated germline with a large 8 cell cyst (dotted white rectangle). (**C**) Bar graph quantifying the number of GSCs per germaria before and one week following irradiation, showing there is no difference in the number of GSCs per germaria. (**D**) Bar graph quantifying the percentage of germaria with four or more progeny, indicating a fully regenerated germline. (**E**) Bar plot showing average p-S6 intensity values for the germaria of 6 week-old w[1118] and nos-Gal4 > *foxo* RNAi flies, showing an increase of p-S6 levels in *foxo* RNAi flies. (**F**) Schematic demonstrating the difference between a wild type aging fly one week following IR and a *foxo* RNAi fly one week following IR. While the wild type fly in incapable of regenerating the germaria, knockdown of *foxo* rescues this defect.

DOI: https://doi.org/10.7554/eLife.27842.014

The following figure supplement is available for figure 8:

**Figure supplement 1.** Foxo represses Tor activity during aging.
DOI: https://doi.org/10.7554/eLife.27842.015

Aging is a complex process, involving the cumulative decline of multiple cell types. Defects in the replicative potential of old GSCs have been reported by other groups (*Zhao et al., 2008*; *Tseng et al., 2014*; *Kao et al., 2015*; *Rauschenbach et al., 2015*). However, our work expands our

understanding of the onset of aging in a unique way. Here, we identify the earliest time point at which defects can be detected in GSC proliferation in an injury model. Before the induction of IR-mediated quiescence in our aging flies, rates of GSC division, as well as the number of GSCs per germaria were similar to that seen in young, healthy flies. Defects were only readily observed following exposure to IR. This suggests that baseline levels of GSC function remain unperturbed, however, the GSCs are unable to recover successfully from insult. This leads us to believe that we have identified a defect early in the initiation of the aging process. Therapeutically, this is a very important window, as it allows us to identify times when an intervention may be useful in helping to slow the progression of aging, or prevent it from initiating in the first place, rather than attempting to reverse it late in the process.

High doses of irradiation have been shown to lead to GSC loss (*Ma et al., 2016*). We specifically utilized a relatively low dose of ionizing radiation, in order to induce damage, but not lead to GSC loss (*Xing et al., 2015*) and to probe the ability of aging stem cells to recover from an injury that should be surmountable were the cells functioning properly. We were able to identify critical roles for two known proteins involved in tissue homeostasis: Foxo for cell cycle withdrawal and Tor for cell cycle reentry. *foxo* has been well documented as a regulator of stem cell self-renewal and quiescence (*Demontis and Perrimon, 2010*; *Xing et al., 2012*; *Eijkelenboom and Burgering, 2013*; *Gopinath et al., 2014*; *Xing et al., 2015*). Notably, *foxo* tends not to be active during normal physiology, but rather during stressful conditions, when it responds to and counteracts a stressor in order to maintain homeostasis (*Kenyon, 2010*; *Eijkelenboom and Burgering, 2013*).

Here, we show that *foxo* activity is required in *Drosophila* GSCs in order for them to withdraw from the cell cycle following exposure to ionizing radiation. There are multiple ways that Foxo may be able to sense the damage caused to the cell by irradiation. In response to the presence of reactive oxygen species, JNK-mediated phosphorylation of Foxo can cause its translocation to the nucleus(*van den Berg and Burgering, 2011*). Foxo can also be the target of multiple pathways that are responsive to DNA damage: Foxo is a target of phosphorylation by ATM (*Matsuoka et al., 2007*) and the MAPK pathway (*Kress et al., 2011*) both of which have been shown to be activated by DNA damage. Lastly, Foxo is capable of directly sensing cellular redox status via oxidation and reduction of amino acids, particularly cysteine (*Dansen et al., 2009*).

CHK2, a highly conserved checkpoint kinase, controls DNA repair, cell cycle arrest and apoptosis following DNA damage. The fly CHK2 ortholog, *loki*, has been shown to mediate GSCs' self-renewal and differentiation following high doses of ionizing radiation (*Ma et al., 2016*). Here we show that depletion of *loki* in the germline prevents GSCs from entering quiescence following exposure to low doses of ionizing radiation. Loki's ability to sense DNA damage and interact with Foxo via the ATM-CHK2-p53 complex (*Chung et al., 2012*) could explain how GSCs know to activate Foxo and withdraw from the cell cycle following IR-induced double stranded breaks. Notably, p53, another component of the ATM-CHK2-p53 complex, has also been shown to regulate GSC irradiation-induced quiescence (*Wylie et al., 2014*) although how p53 interacts with Foxo in this context remains unclear. It is possible that any of these, or the combination of multiple of these systems sense the damage to the GSCs caused by the ionizing radiation and translocate Foxo to the nucleus, initiating IR-induced quiescence.

Mechanistic target of rapamycin (mTOR) signaling has been implicated in a number of

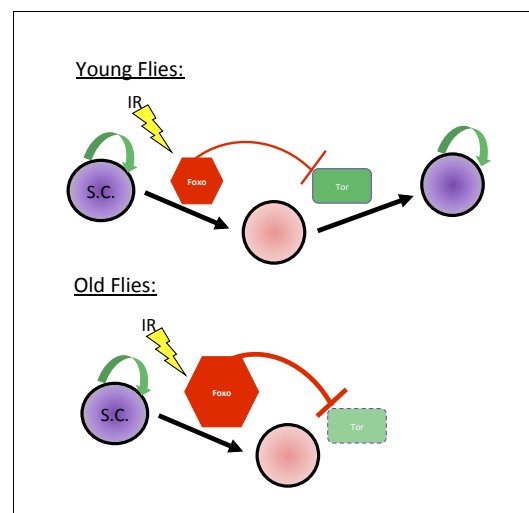

**Figure 9.** Proposed Mechanism. In young flies, injury, such as exposure to ionizing radiation, leads to Foxo activation, which represses *Tor* activity, pushing the GSCs into a state of protective quiescence. Following deactivation of Foxo post-IR, *Tor* activity allows GSCs to reenter the cell cycle and regenerate the germline. In aging flies, increased Foxo activity prevents *Tor* activation and GSCs reentry in the cell cycle.
DOI: https://doi.org/10.7554/eLife.27842.016

different age-related functions, from extension of lifespan (*Vellai et al., 2003*; *Harrison et al., 2009*; *Bjedov et al., 2010*; *Laplante and Sabatini, 2012*; *Bitto et al., 2016*) to germline stem cell self-renewal (*LaFever et al., 2010*; *Sun et al., 2010*), induction of a diapause like quiescent state (*Bulut-Karslioglu et al., 2016*) and muscle satellite cell activation following injury (*Rodgers et al., 2014*). We found that *Tor* signaling was required in order for GSCs to reenter the cell cycle and regenerate the germline following exposure to IR. The sensitivity of wild type GSC proliferation to treatment with rapamycin after IR indicates that this could be mediated via the Tor complex 1 (TORC1) since rapamycin preferentially targets TORC1. We cannot completely rule out a role for Tor complex 2 (TORC2) in GSCs' quiescence since rapamycin treatment has been shown to affect TORC2 activity by keeping Tor associated with TORC1 (*Sarbassov et al., 2006*; *Lamming et al., 2012*). Further studies will focus on investigating the roles of both TORC1 and TORC2 downstream effectors in GSC quiescence.

GSCs with decreased levels of *Tor* activity are unable to reenter the cell cycle post-IR, which is unlikely to be a general consequence of *Tor* inhibition inhibiting GSC division. In a number of different experiments, we observed a more pronounced defect in GSC proliferation in the context of recovery from injury post-IR than at baseline. This indicates that, while *Tor* might play a role in regulating stem cell division and self-renewal under normal physiological conditions, it likely has an additional injury-specific role in helping to replenish adult tissues that have been damaged, either by natural wear and tear or due to disease or injury. Given Tor's ability to regulate translation, nucleotide synthesis, autophagy, lipid synthesis, and proteasome assembly, (*Laplante and Sabatini, 2012*) it will be important to dissect which of these or other cellular processes are required for GSCs' exit from quiescence. It is also quite striking that inhibition of *Tor* resembles the defect observed in aging GSCs, while at an organismal level, inhibition of *Tor* increases lifespan, suggesting a slowing of the aging process. This would indicate that *Tor* inhibition, albeit beneficial at an organismal level, may damage stem cells' capacity to regenerate tissue after injury. This is a particularly important implication of our findings, given the increasing number of anti-aging studies involving rapamycin (*Fan et al., 2015*; *Bitto et al., 2016*).

Mutations in insulin receptor (*InR*) in *Drosophila* and insulin-like growth factor (IGF1) in mice, result in Foxo activation and significant lifespan extension (*Clancy et al., 2001*; *Tatar et al., 2001*; *Bluher et al., 2003*; *Holzenberger et al., 2003*; *Webb et al., 2016*). In humans, single-nucleotide polymorphisms (SNPs) in the FOXO3 locus have been associated with extraordinarily long lifespans (*Morris, 2005*), though the mechanism for this remains elusive. Our study identifies a novel *foxo*-dependent stem cell defect in aged animals in which elevated *foxo* activity prevents GSCs from reentering the cell cycle and regenerating the germline after a challenge. In contrast to other studies showing the benefits of high levels of *foxo* activity, we show, for the first time, that elevated levels of *foxo* activity, albeit beneficial in terms of lifespan extension, are detrimental to stem cell function in the context of tissue regeneration during aging. There are several reasons why pathologically high levels of *foxo* might prevent tissue regeneration in old animals. A meta-analysis of mouse Foxo targets that change with age has revealed that several cell cycle genes, such as the evolutionarily conserved cyclin-dependent kinase 4 (Cdk4), which controls the G1 to S transition, and several ribosomal proteins, which are directly involved in protein translation, are misregulated in aging (*Webb et al., 2016*). In our study, we show how, after IR exposure, *foxo* and *Tor* have opposing patterns of expression in young animals. We also demonstrate how reducing *foxo* levels via RNAi increases p-S6 levels in young and aging animals. This strengthens the idea that *foxo* and *Tor* signaling interact with one another to regulate GSC division following injury and that misregulatin of this crosstalk might contribute to stem cell aging.

Our study shows how Foxo misregulation may impair aging GSCs' regeneration potential. *foxo*'s ability to repress *Tor* could shed light on aging GSCs' inability to resume division following insult. Though the mechanism with which *foxo* and *Tor* interact in the context of aging remains elusive, previous studies have already probed the relationship between these signaling pathways. Foxo has been shown to repress *Tor* signaling by allowing TSC (Tuberous Sclerosis Complex) to localize to the lysosome (*Menon et al., 2014*). At the lysosomal membrane, TSC is then able to inhibit Rheb, an essential activator of mTORC1. Other studies have shown that Foxo is able to inhibit mTORC1 by reducing Raptor levels (*Jia et al., 2004*) or by promoting the transcription of Sestrin 3 and Rictor (*Chen et al., 2010*). Notably, *Tor* signaling can also inhibit Foxo activity by upregulating SGK (*Saxton and Sabatini, 2017*), an AGC-kinase shown to inhibit Foxo. This suggests the possibility of a

negative feedback loop between these signaling pathways. In the future, it will be of vital importance to dissect the crosstalk between *foxo* and *Tor* signaling to understand why GSCs lose their regeneration potential with age.

## Materials and methods

### Fly stocks and culture conditions

The following stocks were obtained from the Bloomington Drosophila Stock Center at Indiana University: w[1118] (RRID:BDSC_3605), P[UAS-Dcr-2.D]1, w1118; P[GAL4-nos.NGT]40 (RRID:BDSC_25751), UASp-*foxo*RNAi (RRID:BDSC_32427 and RRID:BDSC_32993), UASp-*Tor*RNAi (RRID:BDSC_35578), UASp-*Thor*RNAi (RRID:BDSC_36815), UASp-*dm*RNAi (RRID:BDSC_43962), UASp-*dap*RNAi (RRID:BDSC_36720), UASp-*Loki*RNAi (RRID:BDSC_64482). The following stocks were previously generated for and described in *Ward et al., 2006*: UASp-*Delta*/CyO, UASp-*Delta*/CyO; *Dad*-GFP/TM3, UASp-*Delta*/CyO; Ly/TM3. The following stocks were previously generated for and described in *Yu et al., 2009*: pin/CyO;hs-dap-7-7, hsFLP; FRT42B GFP/CyO, FRT42B/CyO, FRT42B *dap*[4]

w[1118] flies were used as a control, unless noted otherwise. Flies were cultured at 25° C on standard cornmeal-yeast-agar medium, augmented with wet yeast. In aging experiments, flies were transferred to fresh vials without wet yeast every 2–3 days. Young and old flies were given wet yeast two days prior to irradiation.

### Gamma-irradiation treatment

After feeding on standard cornmeal-yeast-agar medium augmented with wet yeast paste for two days, young and old flies were transferred to empty vials and treated with 50 Gγs of gamma-irradiation. A Cs-137 Mark I Irradiator was used to administer the proper irradiation dosage, according to instructed dosage chart. Post-treatment animals were transferred back to fresh food with wet yeast and maintained at 25° C until dissection.

### Rapamycin treatment

Following irradiation, flies were place in an empty vial with filter paper soaked in grape juice with either 200 µM rapamycin or DMSO dissolved in it.

### Fertility assay

Following irradiation, 10 females were placed in a new vial with 5 young, unirradiated wild type male flies. Flies were transferred to new vials every 2–3 days and the death of any flies was noted. Vials from flies 5–7 days post-IR were collected and the number of progeny hatched per female was calculated.

### Generation of clones

GSCs clones were induced via the heat shock FLP-FRT system. Young flies (2–3 days old) of the following genotypes *hsFLP; FRT42B GFP/FRT42B, hsFLP; FRT42B GFP/FRT42B dap*[4], were heat shocked in a 37° C water bath for 45 minutes hour once a day for two consecutive days Heat shocked flies were given fresh food and yeast paste every other day until dissection and stored at 25° C for the duration of the experiment.

### Immunocytochemistry

Ovaries were fixed in 4% paraformaldehyde for 15 min, rinsed in PBT (PBS containing 0.2% Triton X-100), and blocked in PBTB (PBT containing 0.2% BSA, 5% normal goat serum) for at least one hour at room temperature. Samples were stored up to 72 hr at 4° in PBTB. The following primary antibodies were used: mouse anti-adducin (RRID:AB_528070 1:30), mouse anti-Lamin C (RRID:AB_528339 1:30) rabbit anti-γH2AV (RRID:AB_828383 1:200), rabbit anti-p-S6 (RRID:AB_916156 1:200), rabbit anti-foxo (generous gift from Pierre Léopold 1:200). Ovaries were incubated with primary antibodies for either 1.5 hr at room temperature or overnight at 4°. After washes with PBT, secondary fluorescence antibodies were utilized including anti-rabbit Alexa 488 (RRID:AB_221544 1:250) and anti-mouse 568 (RRID:AB_2535773 1:250) for 1.5–2 hr at room temperatures in the dark. DAPI was

added to one of the final washes to visualize cells' nuclei. The samples were mounted in glycerol and analyzed on a Leica SPE5 confocal laser-scanning microscope.

## Statistical analysis

All data are presented as the mean of at least three independent experiments ($n \geq 3$) with the standard error of the mean (SEM) indicated by error bars, unless otherwise indicated. Statistical significance was determined using Student's t test (for two groups) or ANOVA with the appropriate post hoc test (for more than two groups). Data were compiled using Excel 2013 software and analyzed using Excel (version 2013 for Windows; Microsoft, Seattle, WA, USA) or the Astatsa Online Web Statistical Calculator (astatsa.com, Philadelpha, PA, USA).

## Acknowledgements

We thank the TRiP at Harvard Medical School (NIH/NIGMS R01-GM084947) for providing transgenic RNAi fly stocks used in this study. We would like to thank the members of the Ruohola-Baker lab for their stimulating discussion and valuable comments. We thank Sonthaya Artphakdi for help with the nos-Gal4 > Delta studies. This work is supported by a Genetic Approaches to Aging Training Grant postdoctoral fellowship from the National Institute of Aging to REK, gift from Hahn Family and partly by grants from the National Institutes of Health R01GM097372, R01GM083867, 1P01GM081619, U01HL099997; UO1HL099993 for HRB.

## Additional information

### Funding

| Funder | Grant reference number | Author |
|---|---|---|
| National Institute on Aging | Genetic Approaches to Aging Training Grant postdoctoral fellowship | Rebecca E Kreipke |
| Coordenação de Aperfeiçoamento de Pessoal de Nível Superior | Science without Borders | Ondina Palmeira |
| National Institute of General Medical Sciences | R01-GM084947 | Hannele Ruohola-Baker |
| Hahn Family | | Hannele Ruohola-Baker |
| National Institutes of Health | UO1HL099993 | Hannele Ruohola-Baker |
| National Institutes of Health | R01GM097372 | Hannele Ruohola-Baker |
| National Institutes of Health | R01GM083867 | Hannele Ruohola-Baker |
| National Institutes of Health | 1P01GM081619 | Hannele Ruohola-Baker |
| National Institutes of Health | U01HL099997 | Hannele Ruohola-Baker |

The funders had no role in study design, data collection and interpretation, or the decision to submit the work for publication.

### Author contributions

Filippo Artoni, Conceptualization, Data curation, Formal analysis, Validation, Investigation, Visualization, Methodology, Writing—original draft, Writing—review and editing; Rebecca E Kreipke, Conceptualization, Data curation, Formal analysis, Funding acquisition, Validation, Investigation, Visualization, Methodology, Writing—original draft, Writing—review and editing; Ondina Palmeira, Formal analysis, Investigation, Visualization, Methodology; Connor Dixon, Formal analysis, Investigation, Visualization; Zachary Goldberg, Resources; Hannele Ruohola-Baker, Conceptualization, Supervision, Funding acquisition, Writing—original draft, Writing—review and editing

## Author ORCIDs

Filippo Artoni (iD) http://orcid.org/0000-0003-3902-9217
Rebecca E Kreipke (iD) https://orcid.org/0000-0003-3563-3054
Hannele Ruohola-Baker (iD) http://orcid.org/0000-0002-5588-4531

## Decision letter and Author response

Decision letter https://doi.org/10.7554/eLife.27842.021
Author response https://doi.org/10.7554/eLife.27842.022

## Additional files

### Supplementary files

• Source data 1. Excel file compiling source data for the most relevant experiments. Data are arranged by type of experiment and then further organized by genotypes, time points and replicate number.
DOI: https://doi.org/10.7554/eLife.27842.017

• Supplementary file 1. (A) Table showing the number of GSCs with elongated spectrosomes expressed as mean percent values (±SE). (B) Table showing the number of germaria with branched fusomes expressed as mean percent values (±SE).
DOI: https://doi.org/10.7554/eLife.27842.018

• Transparent reporting form
DOI: https://doi.org/10.7554/eLife.27842.019

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
