## [Decision Letter]

Thank you for submitting your article "Loss of FOXO rescues stem cell aging in *Drosophila* germline stem cells" for consideration by *eLife*. Your article has been reviewed by two peer reviewers, and the evaluation has been overseen by a Fiona Watt as the Senior Editor and Reviewing Editor. The reviewers have opted to remain anonymous.

The reviewers have discussed the reviews with one another and the Reviewing Editor has drafted this decision to help you prepare a revised submission.

Summary:

Kreipke et al. examine the mechanism by which exposure of germline stem cells (GSC) to ionizing radiation (IR) leads to a transient block in cell cycle progression followed by subsequent regeneration of the germline in young fruit flies. In contrast IR triggers a permanent inhibition of germline production in older flies but does not eliminate the GSC, suggesting that there is some form of regulation of exit and re-entry of the cell cycle for young but not older germlines. Using a combination of RNAi knock-downs and histochemical staining the authors analyze the mechanistic basis for the regulation of the GSC entry/exit from the cell cycle and find that the stress induced transcription factor Foxo and the metabolic homeostasis/aging pathway regulator Tor play key opposing roles in this process. Following IR exposure Foxo levels increase and inhibit Tor signaling. Tor signaling is required for post-IR treated quiescent GSCs re-entering the cell cycle, while Foxo induces exit into the quiescent state. Perhaps the most interesting result, which has potential practical implications for combatting Tor-dependent aging, is that RNAi knock-down of Foxo can rescue quiescent GSC in older flies resulting in regeneration of a germline. These conclusions are well supported by high quality data and should be of broad interest to readers in the fields of stem cell biology and aging.

Essential revisions:

Below are listed minor revisions that the reviewers request in order to improve the clarity of the paper.

1) In the second paragraph of the subsection “Aging germline stem cells survive exposure to ionizing radiation, but fail to re-enter the cell cycle in a timely fashion”, the authors indicate that the majority of germaria still had GSCs 7 days after irradiation. This is correct but at both 4 and 6 weeks a decrease in numbers of GSCs is evident. Can the authors please comment?

2) In Figure 1 the authors show a decrease in germaria with >4 progeny. Why is this not 100% in unirradiated germaria?

3) The loss of GSCs after irradiation in germaria with ectopic Delta (Figure 3) should be quantified.

4) It would be informative if Tie activation could be shown only in cells that border daughters in Figure 3 (if such an assay is possible).

5) The way data are displayed in Figure 5 does not provide an opportunity for the reader to determine the decrease in WT GSCs with elongated spectrosomes – how can the reader compare foxoRNAi to WT to determine that foxoRNAi GSCs keep dividing at a normal rate? This could be fixed by presenting scatter plots with mean values.

6) The increase in Foxo staining in Figure 8 appears to be mainly in 8-16 cell cysts. Is there an increase in GSCs?

7) The increase in p-S6 staining in Figure 8 appears slight and would benefit from quantification.

8) In Figure 8, panels C and D it would be helpful to include controls for foxo>RNAi without GAL4 for comparison.

9) In Figure 1—figure supplement 1 the authors show a very simple but compelling phenotype in which old IR treated females had virtually no offspring as judged by the generation of pupa. Did the authors observe rescue of this phenotype by Foxo RNAi? If so, perhaps they could show that result and if not, perhaps they could mention this fact and discuss possible reasons for why not (e.g., other effectors might also be essential for full germline regeneration?)

Further points:

1) Figure 7 shows baseline expression not 2 days post irradiation as indicated in the subsection “Foxo represses Tor in GSC after IR”.

2) Foxo staining should always be referred as "Foxo" not "foxo".

---

## [Author Response]

Essential revisions:1) In the second paragraph of the subsection “Aging germline stem cells survive exposure to ionizing radiation, but fail to re-enter the cell cycle in a timely fashion”, the authors indicate that the majority of germaria still had GSCs 7 days after irradiation. This is correct but at both 4 and 6 weeks a decrease in numbers of GSCs is evident. Can the authors please comment?

As previously described, *Drosophila* females will lose GSCs during late stages of aging process. The small loss of GSCs observed at 4 and 6 weeks is an early manifestation of this process.

2) In Figure 1 the authors show a decrease in germaria with >4 progeny. Why is this not 100% in unirradiated germaria?

The decrease in germaria with >4 progeny reflects the small loss in GSC number observed in aging animals.

3) The loss of GSCs after irradiation in germaria with ectopic Delta (Figure 3) should be quantified.

We have now quantified GSCs loss 1 day following IR in germaria with ectopic Δ and added the information in Figure 3—figure supplement 1.

4) It would be informative if Tie activation could be shown only in cells that border daughters in Figure 3 (if such an assay is possible).

While this is an important question, at this time Tie activation is difficult to show directly since no reliable *Drosophila* P-Tie antibody exists to our knowledge.

5) The way data are displayed in Figure 5 does not provide an opportunity for the reader to determine the decrease in WT GSCs with elongated spectrosomes – how can the reader compare foxoRNAi to WT to determine that foxoRNAi GSCs keep dividing at a normal rate? This could be fixed by presenting scatter plots with mean values.

To provide the opportunity for the reader to determine the decrease in WT GSC with elongated spectrosomes we have now included the data used for the graph in Supplementary file 1.

6) The increase in Foxo staining in Figure 8 appears to be mainly in 8-16 cell cysts. Is there an increase in GSCs?

We show a high magnification image of 6-week old GSCs stained for Foxo (see Author response image 1). However, since we were not able to show that the difference is significant, we excluded the statement from the manuscript.

7) The increase in p-S6 staining in Figure 8 appears slight and would benefit from quantification.

We have now quantified the P-S6 increase in FOXO KD vs. control in aged animals. The increase of P-S6 in FOXO KD animals is significant, suggesting that FOXO based aging phenotype manifests through Tor and its target P-S6.

8) In Figure 8, panels C and D it would be helpful to include controls for foxo>RNAi without GAL4 for comparison.

We have now analyzed foxo>RNAi without GAL4. No obvious defects were observed (see Supplementary file 1).

9) In Figure 1—figure supplement 1 the authors show a very simple but compelling phenotype in which old IR treated females had virtually no offspring as judged by the generation of pupa. Did the authors observe rescue of this phenotype by Foxo RNAi? If so, perhaps they could show that result and if not, perhaps they could mention this fact and discuss possible reasons for why not (e.g., other effectors might also be essential for full germline regeneration?)

We quantified the offspring of 6-week old Foxo RNAi flies before and after IR. We did not observe a rescue of the offspring phenotype in wild type 6-week old flies. As suggested by the reviewer, other effectors might also be essential for full germline regeneration.

Further points:1) Figure 7 shows baseline expression not 2 days post irradiation as indicated in the subsection “Foxo represses Tor in GSC after IR”.

We have now corrected this statement in the subsection “foxo represses Tor in GSC after IR”.

2) Foxo staining should always be referred as "Foxo" not "foxo".

We have now corrected this.